# Protecting the public interest while regulating health professionals providing virtual care: A scoping review

**Kathleen Leslie**[1]*, **Sophia Myles**[1,2], **Catharine J. Schiller**[3], **Abeer A. Alraja**[1], **Sioban Nelson**[2‡], **Tracey L. Adams**[4‡]

**1** Athabasca University, Athabasca, Alberta, Canada, **2** University of Toronto, Toronto, Ontario, Canada, **3** University of Northern British Columbia, Prince George, British Columbia, Canada, **4** Western University, London, Ontario, Canada

‡ SN and TLA are joint senior authors on this work.
* kleslie@athabascau.ca

**Data Availability Statement:** All relevant data are within the manuscript. Further data that supports the findings can be found in the open-access report for this project at https://www.sshrc-crsh.gc.ca/

## Abstract

Technology is transforming service delivery in many health professions, particularly with the rapid shift to virtual care during the COVID-19 pandemic. Health profession regulators must navigate legal and ethical complexities to facilitate virtual care while upholding their mandate to protect the public interest. The objectives of this scoping review were to examine how the public interest is protected when regulating health professionals who provide virtual care, discuss policy and practice implications of virtual care, and make recommendations for future research. We searched six multidisciplinary databases for academic literature published in English between January 2015 and May 2021. We also searched specific databases and websites for relevant grey literature. After screening, 59 academic articles and 18 grey literature sources were included for analysis. We identified five key findings: the public interest when regulating health professionals providing virtual care was only implicitly considered in most of the literature; when the public interest was discussed, the dimension of access was emphasized; criticism in the literature focused on social ideologies driving regulation that may inhibit more widespread use of virtual care; subnational licensure was viewed as a barrier; and the demand for virtual care during COVID-19 catalyzed licensure and scope of practice changes. Overall, virtual care introduces new areas of risk, potential harm, and inequity that health profession regulators need to address as technology continues to evolve. Regulators have an essential role in providing clear standards and guidelines around virtual care, including what is required for competent practice. There are indications that the public interest concept is evolving in relation to virtual care as regulators continue to balance public safety, equitable access to services, and economic competitiveness.

## Author summary

Technology is transforming how many health professionals provide services, particularly with the rapid shift to virtual care during the COVID-19 pandemic. Many of these health

society-societe/community-communite/ifca-iac/
evidence_briefs-donnees_probantes/skills_work_
digital_economy-competences_travail_economie_
numerique/leslie-eng.aspx.

**Funding:** This project was supported by a grant co-funded by the Social Sciences and Humanities Research Council (https://www.sshrc-crsh.gc.ca/home-accueil-eng.aspx) and the Government of Canada's Future Skills Centre (https://fsc-ccf.ca/) (#872-2020-0022 to KL). The funders had no role in study design, data collection and analysis, decision to publish, or preparation of the manuscript.

**Competing interests:** The authors have declared that no competing interests exist.

professionals are accountable to a regulator that sets standards of practice, including for virtual care. These regulators have a mandate to protect the public. We conducted a review to determine whether there was existing evidence or literature about how these regulators were working to protect patients when health professionals were providing virtual care. Most of the literature we found did not explicitly discuss the public interest in regulating health professionals who provide virtual care. However, when the public interest was discussed, access to care was emphasized. Criticism in the literature focused on social ideologies driving regulation that may inhibit more widespread use of virtual care, especially as the demand for virtual care during COVID-19 catalyzed regulatory changes. Virtual care introduces new areas of risk, potential harm, and inequity that regulators need to address as technology continues to evolve. Regulators have an essential role in providing clear standards and guidelines around virtual care, including what is required for health professionals to be competent.

## Introduction

Technology is increasingly transforming service delivery in the health professions. Health profession regulators face legal and ethical complexities in facilitating access to safe, high-quality virtual care while continuing to protect the public interest. The rapid shift to virtual care during the COVID-19 pandemic [1–3] made the need to reform regulatory practices more urgent. Professional regulators are legally obligated to protect the public interest, but how the public interest is defined is amorphous and subject to social change [4], and is also closely tied to historical practice and population health needs specific to the jurisdiction (state/province, national, or supra-national) in which regulation takes place. The emergence of the global digital economy has highlighted the complexities involved in public interest regulation in the context of exponential technological advances and their application in professionals' workplaces.

The paramountcy of the public interest is often enshrined in legislative frameworks, requiring regulators to protect the public against negligence, dishonesty, and incompetence by ensuring only those fit to practice safely are registered or licensed to practice [5,6]. Professional regulators typically aim to meet this public protection mandate by setting entry-to-practice standards, maintaining a register of licensed practitioners, and monitoring and enforcing conduct, competency, and capacity in practice [6]. Typically, this involves some form of title protection (referred to as registration in some jurisdictions) or reservation of practice (referred to as licensing in some jurisdictions) [7]. While there are other frameworks for assuring the safety and quality of the services provided by health practitioners, including voluntary certification by professional associations, negative licensing, and accredited registers [7], we focused in this review on those practitioner groups regulated by a body with a statutory mandate to protect the public. While this type of regulation is most prominent in high-income Anglophone jurisdictions with a common law history, many low- and middle-income nations also regulate in the public interest, though with resource constraints that may limit the full spectrum of regulatory functions and activities [8,9].

Despite the mandate to protect the public, professional regulation has historically been criticized as potentially being driven by private interests since it can result in reduced competition for services and actions that benefit the professionals being regulated, at the expense of the public who must pay more for their services [10,11]. The long-recognized theory that regulators become captured by vested interests of the industry being regulated [12] continues to drive regulatory scrutiny and reform [13,14]. In particular, regulatory restrictions around

scope of practice have been criticized for being anti-competitive and not reflecting the public interest [15,16].

The social construct of the public interest will remain in flux in the digital era. Some regulators have altered their activities or policies to include the following: ensuring that standards provide necessary guidance for the impact of new models of digital work on topics such as consent, documentation, records management, privacy, and confidentiality [17,18]; facilitating inter-jurisdictional virtual work [19,20]; and adapting continuing competence requirements and disciplinary procedures to reflect modern digital environments [21]. Regulators must respond to new technologies because these technologies have the potential to significantly alter professional practice and relationships between practitioners and clients. Moreover, regulation needs to align with new sociotechnical landscapes, including possible negative features such as new or exacerbated potential risks, inequities, and market failures [22]. Thus, ensuring effective regulation is less about the technology itself than the impact on society when such technology is used.

Participation in virtual care for many health professions is not new [20] and has long been used to improve the availability and accessibility of healthcare in rural areas. However, the COVID-19 pandemic rapidly accelerated the pivot to virtual care [1–3]. The exponential growth of virtual care distinguishes the response to COVID-19 from previous public health crises or emergencies (e.g., SARS, H1N1). Reforms to regulate virtual practice were similarly fast-tracked, and calls have been made for regulatory frameworks to undergo careful post-implementation reviews to ensure public protection [23]. Others have argued that relaxing regulatory barriers during the pandemic demonstrates that these requirements were unnecessary for public protection and have called for changes to be retained post-pandemic [24,25]. Regulatory changes required to facilitate virtual work (either during steady state or crisis situations) are beginning to receive academic attention, particularly in federated jurisdictions that regulate health professionals subnationally [26–28].

## Review question and objectives

We conducted a scoping review to generate a broad overview of this topic and summarize the available evidence. We conducted a preliminary search of PROSPERO, MEDLINE, the Cochrane Database of Systematic Reviews, and JBI Evidence Synthesis in 2021 and found no scoping or systematic reviews on this topic. This scoping review is part of a larger project on regulating virtual work that reviewed additional sources for professions outside health and presented policy case studies to provide a deeper dive into specific Canadian examples and contexts. The report for the full project is openly available on Athabasca University's institutional repository [29], and the evidence brief is available on the website of the funding agency, the Social Sciences and Humanities Research Council [30].

The research question for this scoping review was: *How is the public interest protected when regulating health professionals engaged in virtual care*? The objectives of this scoping review were to examine how the public interest is protected when regulating health professionals providing virtual care, to discuss policy and practice implications related to professional regulation of virtual care, and to make recommendations for future research.

## Methods

We conducted this scoping review following the JBI methodology for scoping reviews and in line with the Preferred Reporting Items for Systematic Reviews and Meta-Analyses extension for Scoping Reviews PRISMA-ScR checklist (see S1 PRISMA-ScR checklist) [31,32]. Our

scoping review protocol was registered with the Open Science Framework (https://doi.org/10.21203/rs.3.rs-789608/v1) and published in a peer-reviewed journal [33].

## Inclusion criteria

We used the *Population-Concept-Context* approach recommended by the JBI scoping review methodology to define our inclusion criteria. Our *population* included any studies involving regulated health professionals, defined for the purposes of our review as those governed by professional regulators with the legal mandate to protect the public. Studies that discussed the *concept* of protecting the public interest (in the narrower sense of ensuring patient/client safety or the broader sense of acting in the interests of the public) when regulating health professionals providing virtual care were considered for inclusion. In searching for literature on virtual care, we included any form of health professional service delivery without in-person interaction. We considered studies from any jurisdictional *context* if regulatory activities relevant to professionals providing virtual care were discussed. Articles that did not discuss regulation by professional regulators with a public protection mandate (e.g., those focused on other aspects of health sector regulation such as pharmaceuticals, medical devices, or provider reimbursement; those focused on voluntary certification by professional associations or accredited registries) were excluded.

## Types of sources

English-language academic literature incorporating any type of study design published between January 2015 and May 2021 was considered for inclusion. English-language grey literature in the public domain (such as government reports, documents from regulatory consortiums, and policy papers) that specifically discussed regulating health professionals providing virtual care was also considered for inclusion. Letters to the editors, textbook chapters, and conference abstracts were excluded. We also excluded practice guidance or standards from individual health profession regulators (given the volume of documents this would represent).

## Search strategy

An initial limited search of the Scopus database was conducted to identify relevant articles on the topic. The words in the titles and abstracts of these articles, as well as the index terms, were used to develop a full search strategy (see S1 File). This search strategy was then adapted across the other databases in consultation with a research librarian.

The literature on health profession regulation of virtual practice is diverse, and we wanted to capture various disciplinary perspectives. Therefore, we searched six multidisciplinary academic databases (MEDLINE [Ovid], Embase [Ovid]), Sociological Abstracts [Proquest], Social Work Abstracts [EBSCO], Scopus and Google Scholar). Additionally, an iterative grey literature search was conducted, including grey literature databases (OpenGrey, Nexis Uni, ProQuest Dissertations and Theses Global), search engines (first 200 Google results), and relevant websites (e.g., OECD Regulatory Policy Division, GovInfo, and Government of Canada) based on keywords used for the academic databases and in consultation with the research team. Studies published in English from 1 January 2015 until 29 May 2021 (when the searches were conducted) were included. This date range was chosen for feasibility given the rapidly evolving nature of virtual care, particularly during the COVID-19 pandemic. Reference lists of articles selected for inclusion were screened for additional papers not captured in the database searches.

## Study selection

Following the search, all identified records were collated and uploaded into the review management software Covidence and duplicates were removed. Two reviewers independently screened all titles and abstracts against the inclusion criteria for the review. Potentially relevant papers were retrieved in full; two reviewers assessed full-text studies and excluded those that did not meet the inclusion criteria. Reasons for exclusion at this stage are reported in the PRISMA diagram (see Fig 1). Any disagreements between reviewers were resolved through discussion or by a third reviewer to achieve consensus.

## Data extraction and analysis

Data were extracted from included papers by one reviewer and verified by a second reviewer using a modification of the JBI data extraction tool developed for this scoping review by the research team (available as a supplementary file with our study protocol). Data extraction tools for academic and grey literature were pilot tested on 10 sources and revised to ensure that the most relevant information was captured appropriately. The data extracted included details about the *population* (e.g., the specific health profession), the *concept* of regulating health

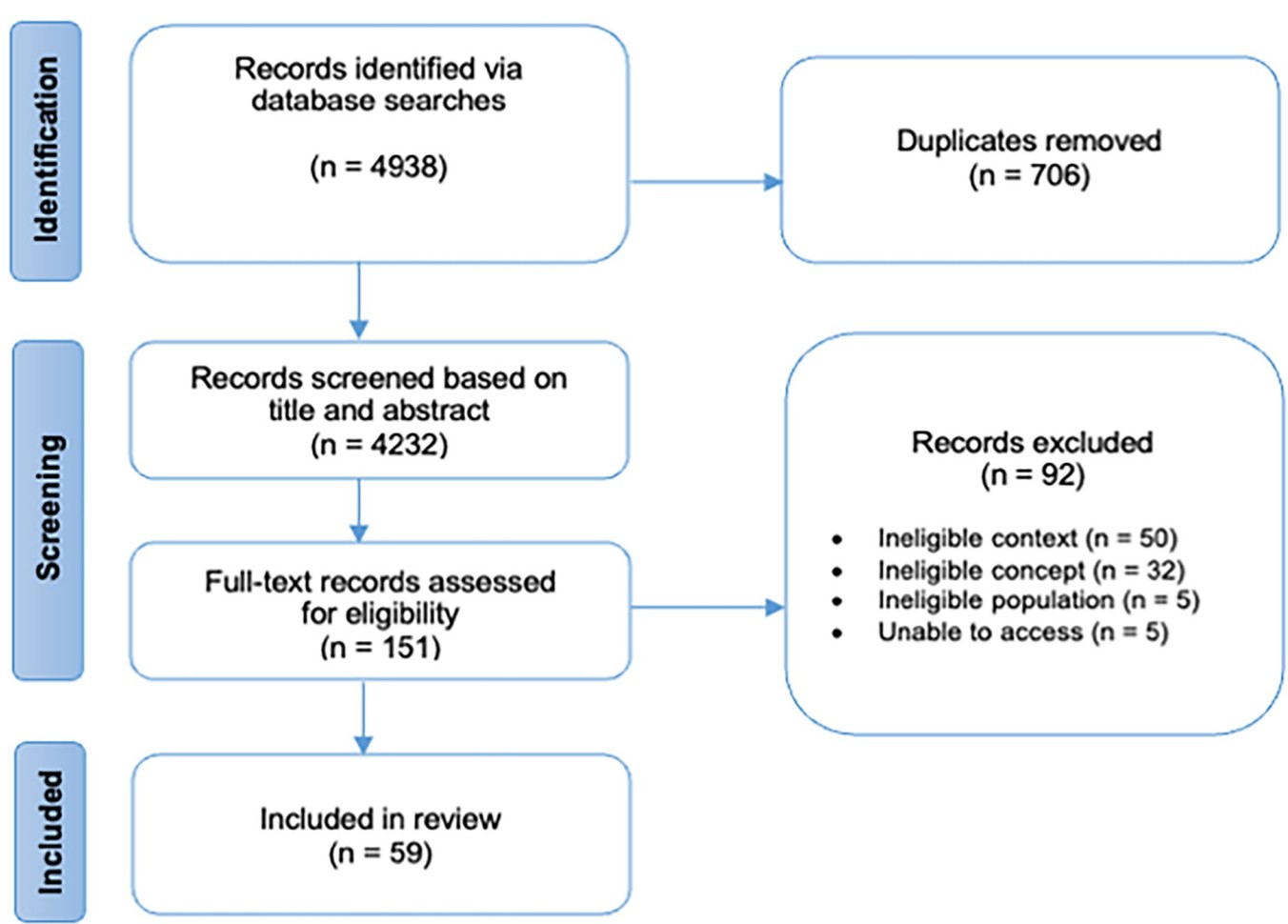

**Fig 1. Search results and study selection and inclusion process [34].**

professionals providing virtual care, the *context* of regulatory activities in the public interest, type of article, geographic location, and key findings relevant to the review objectives and question. Analysis of included studies was conducted by summarizing these characteristics descriptively and synthesizing the extracted data according to key findings. Based on our review objectives, we have highlighted implications for regulatory policy and professional practice and knowledge gaps warranting further research.

## Results

The academic database searches resulted in 4232 records after the duplicates were removed. After screening titles and abstracts, 151 full-text papers were assessed for eligibility, 92 of which were excluded based on our inclusion criteria. A total of 59 academic articles were included in our final review. The PRISMA flow diagram for the academic literature is presented in Fig 1.

### Characteristics of included articles

Academic literature included in this review consisted of the following types of papers: opinions/commentaries/perspectives/debates/expert reports (n = 24); reviews including meta-syntheses, narrative reviews, literature reviews and scoping reviews (n = 13); legal research notes/analyses/reviews (n = 4); descriptive/observational studies (n = 3); qualitative studies (n = 2); case reports/studies (n = 1), and other article types (n = 8).

Geographically, most papers focused on the United States (n = 44). The geographic contexts of the remaining articles were Canada (n = 2), India (n = 1), Hong Kong (n = 1), Brazil (n = 1), South East Asia (n = 1), as well as articles with a multi-country focus including Europe or the European Union (EU) (n = 7), the EU and Australia (n = 1), and Australia, Portugal, and Russia (n = 1).

Most articles focused on physicians (n = 34) and medical specialties or services, including radiology, psychiatry, surgery, dermatology/dermatopathology, and doctors providing abortions by virtual care. Professional populations highlighted in the remaining articles were nurses (registered nurses, nurse practitioners, advanced practice registered nurses); dentists, oral and maxillofacial radiologists, and allied dental providers (i.e., dental therapists); pharmacists and pharmacy technicians; allied health professionals (e.g., occupational therapists, physical and physiotherapists, speech-language pathologists), and mental health professionals (e.g., psychologists, social workers). A few articles discussed regulated health providers more generally.

### Varying terminology used in the literature for virtual care

The literature contained varying terms and definitions to interpret aspects of virtual care, such as remote services, telehealth, and telemedicine. These terms and associated definitions varied based on a combination of profession or specialty, service(s) provided, technology and tools used to provide services, client population(s) served, location(s) of care, and proposed new specializations (i.e., medical virtualist). We have summarized the varying terms encountered in the literature in Table 1. Telemedicine and telehealth were most frequently used in the academic literature to discuss virtual care when regulating health professions. In this paper, we use the term virtual care in line with the recent movement by Canadian medical regulators to adopt this term in practice standards and other regulatory documents (see, e.g., the Federation of Medical Regulatory Authorities of Canada [FMRAC] Virtual Care Framework) [21].

**Table 1. Examples of terms used in the academic literature to describe practice encounters not involving in-person contact.**

| | | |
|---|---|---|
| Connected care | Tele-coaching | Telepathology |
| Cybermedicine | Telecollaboration | Telepharmacy |
| Digital health | Teleconsulting | Telephone nursing |
| Digital technologies | Telecommunication | Telephonic care |
| Digital tools | Teledentistry | Telepsychiatry |
| e-connected care | Teledermatology | Telepsychology |
| eHealth | Teledermatopathology | Teleradiology |
| Geriatric telemedicine | Telediagnosis/Telediagnostic | Telerehabilitation |
| Information and communications technology | Tele-education | Tele-support |
| Information technology | Telehomecare | Telesurgery |
| MHealth | Telehealth | Teletreatment |
| Remote consultation | TeleICU | Televisit |
| Remote intervention | Telemedicine | Virtual care |
| Remote patient monitoring | Telemedicine abortion | Virtual consultation |
| Telecardiology | Telenursing | Virtual presence |
| Telecare | Telemental health | Virtual services |
| | Teleneurology | Virtual technology |

## Key findings

We identified five key findings in our evidence synthesis of the academic and grey literature on this topic.

### 1. The public interest when regulating health professionals providing virtual care was only implicitly considered in most of the literature

The expansion of virtual service provision raises many challenges for regulators. The literature we reviewed, however, did not always specifically articulate public interest concerns in regulating virtual care. In some sources, public interest considerations were implied by the regulatory activities discussed, such as the public health impacts of relaxed prescribing rules and preventing incompetent or unethical practice when services are provided virtually [35–42]. The literature highlighted several virtual care concerns that impact the public interest created by current regulatory structures. Articles also used terms such as protecting (or enhancing) public safety, health, and welfare in the context of regulating professional practice.

The literature primarily discussed licensure as a regulatory activity when discussing virtual practice (n = 45). Other regulatory activities described in the literature included certification, credentialing and privileging; the issuance of emergency orders and waivers during the COVID-19 pandemic; scope of practice; practice standards (including ethics and continuing competence); standards of care; duty to care; and guidance around issues such as requirements to consult with other providers and how to ensure privacy, security, and confidentiality when providing virtual care. Some articles identified that regulators needed to clarify specific aspects of virtual care for health professionals, including clarifying the expected standards for virtual care, determining when a practitioner-client relationship is established, and managing ethical issues [43–51]. Recognizing this need, umbrella regulatory consortiums have offered guidance, model codes, and support for memoranda or agreements for virtual care practice standards. In Canada, the Canadian Alliance of Physiotherapy Regulators established a cross-border physiotherapy memorandum of understanding to facilitate access to telerehabilitation [52]. The Federation of Medical Regulatory Authorities of Canada, at the time of this review, was exploring the potential for a license specific to virtual care, a short-duration license for portability, or a fast-tracked license agreement for physicians already licensed in another Canadian jurisdiction [53]. In the United States, regulatory umbrella organizations such as the National Council of

State Boards of Nursing and the Federation of State Medical Boards have been instrumental in the proliferation of interstate licensure compacts for their respective professions. These compacts overcome the barrier of needing multiple licenses to provide services across state borders by allowing providers to practice in other compact states where they do not hold a license as long as they hold a license in good standing in a home state that is a compact member [54,55]. While not explicitly or specifically stated in the reviewed literature, the above examples suggest implicit discussion of the public interest as central to regulatory functions, activities, and decisions around virtual care, in line with jurisdictionally-specific and socially-influenced constructions of the public interest.

### 2. The dimension of access was emphasized when regulating health professionals providing virtual care

When the public interest in virtual care was discussed in the literature, the dimension of access to affordable services was given the most weight. Indeed, virtual care has traditionally been used to facilitate more equitable access to healthcare in rural and remote areas to help address health disparities and health workforce shortages [56,57]. Increasing healthcare access in underserviced areas is thought to improve public health and welfare [57]. As virtual care continues to be commonplace in pandemic recovery and the post-pandemic world, a challenge for professional regulators will be to continue to adapt as virtual care evolves to ensure patient access to safe and competent care provision. There is also a growing public expectation of access to virtual care. The Organization for Economic Co-operation and Development has noted that societies across the globe are adopting digital technologies, and consumers are increasingly expecting the same level of responsiveness and ease of use in healthcare as in other digital technologies in other areas of life [58].

The extent to which professional regulation is fulfilling its mandate to act in the public interest as it pertains to virtual care was questioned in the literature. In the American context, legal and regulatory mechanisms were criticized as impeding healthcare access by potentially restricting some providers from delivering medically necessary services [40]. For example, current Interstate Medical Licensure Compact standards may exclude approximately 20% of physicians from being eligible for compact licensure [43]. This is partly attributable to restrictive eligibility requirements (e.g., primary residence and at least 25% of medical practice must occur in a declared state of principal license) and individual practitioner self-determinations of eligibility to apply to participate in the compact [59].

One somewhat opposing concern raised in the literature was the potential for increased access to addictive or illegal drugs because of relaxed teleprescribing for controlled substances that did not require conducting an in-person examination [35]. Several articles highlighted court cases and resulting legislation (*Ryan Haight Online Pharmacy Consumer Protection Act*, 2008; *Hageseth v Superior Court*, 2007) regarding online prescribing of controlled substances or other medications without an initial in-person exam [37,47,60,61]. Before the COVID-19 pandemic, Australia, South Africa, Spain, and the UK had uniform rules for in-person and teleprescribing. While restrictions on teleprescribing in many jurisdictions were relaxed during the pandemic to increase service access, there were still restrictions on prescribing medications via telepsychiatry in some jurisdictions [62]. While regulators understandably have quality and safety concerns for controlled drug prescriptions and psychiatric care delivered by virtual care, evidence regarding the extent to which these concerns are founded is mixed. Some articles we reviewed emphasized the mixed evidence around whether overprescribing occurred when clinical encounters did not involve in-person contact [49,63].

### 3. Criticism focused on social ideologies driving regulation that may inhibit more widespread use of virtual care

The reviewed literature questioned whether regulatory decisions are in the public interest and identified criticisms about current regulatory structures that could inhibit more widespread use of virtual care. These criticisms around virtual care regulatory policy fell into two broad categories: that these policies potentially limited competition and created economic inefficiencies (i.e., health as a competitive business), and that they were driven by politics rather than sound evidence on the appropriateness of virtual care (i.e., ideological views on 'controversial care').

Requiring patients or clients to travel for services that can be safely provided virtually could be viewed as acting in the interests of professionals to reduce competition for local providers. Many states limit virtual care provision by requiring health providers to be licensed in the state where the client resides [49,64]. Practice limitations like this may be subject to federal antitrust (competition law) challenges. The papers we reviewed highlighted two antitrust cases in the United States that involved health profession regulators. In *Teladoc v Texas Board of Medicine*, a national virtual care platform sued the Texas Board of Medicine for mandating that consultations be conducted face-to-face. Teladoc dropped its antitrust lawsuit after the Texas Board of Medicine enacted new regulations allowing virtual care without a prior face-to-face interaction, paving the way for increased virtual care provision and potentially driving competition for local in-person practitioners [65,66]. In *North Carolina State Board of Dental Examiners v Federal Trade Commission*, the Supreme Court of the United States determined that licensing boards could only claim to be immune from federal antitrust actions if those boards were under active state supervision. In short, while licensing boards must protect the public interest, they need to do so in a way that balances public safety with promoting a robust, competitive economy [67]. Competition and politics in health have a different meaning in the American context and may not be similarly prioritized in other countries, particularly given the framing of health as a predominantly private good in the United States.

Similarly, while registration or licensing fees are required to support regulatory programs and activities, requiring professionals to pay licensing fees in multiple jurisdictions potentially creates undue time and financial burdens to provide telemedicine services. Obtaining and maintaining licenses for multiple jurisdictions is costly and time-consuming for providers [68,69] and arguably raises costs for consumers, thereby impacting equitable access to care [70].

Restrictions on politically controversial services such as abortion via virtual care were criticized as being motivated by politics rather than science, thus resulting in questionable policies promulgated by medical boards [71]. The example most commonly discussed in the literature we reviewed was a practice standard instituted by the Iowa Board of Medicine prohibiting abortion by virtual care. In a legal challenge brought by Planned Parenthood, the Iowa Supreme Court found that the medical board had singled out medical abortion despite evidence that the number of abortions and adverse outcomes do not increase when this service is provided by virtual care [72]. In Canada, there have not been similar assertions of regulatory restrictions on politically controversial services by virtual care. A protocol released early in the pandemic by the Society of Obstetricians and Gynecologists of Canada provided guidance on telemedicine abortion in Canada and did not suggest any in-person visit requirements [73]. Again, this may relate to the framing of health services in Canada, where universal health coverage exists, compared to the United States. Note that our review was conducted before the 2022 United States Supreme Court decision in *Dobbs v. Jackson Women's Health Organization* (discussed below).

**4. Subnational occupational licensure was viewed as a barrier to virtual care**

Critical to regulating virtual practice is the issue of cross-jurisdictional practice in federated jurisdictions such as the United States and Canada where health profession regulation initially emerged at the subnational (state/provincial/territorial) level to govern services provided locally, address regional concerns, and reflect local circumstances. However, technologies now facilitate cross-regional practice and the highly variable geographic approach to licensure and registration has become increasingly problematic and difficult to defend. The literature identified regulation at the subnational level in Canada and the United States as a barrier to equitable access to professional services, particularly during the COVID-19 pandemic. This was the most consistent point raised across the included articles and reflects the dominance of these countries in the literature we reviewed. Cross-jurisdictional virtual care is much less of a concern in countries where regulation occurs nationally (e.g., New Zealand), though an argument could be made about the need to regulate virtual care more globally, particularly for future conflict or crisis emergency situations.

In Canada, where the constitutional division of powers has provided provinces and territories authority to regulate most professions, calls have been made to address the variability in the approach to professional regulation across provincial and territorial borders [74,75]. Despite some regulatory reform around virtual care, variations in licensure requirements and scopes of practice, as well as difficulties ascertaining to which regulator professionals are accountable, have continued to complicate virtual cross-jurisdictional practice [53]. A pilot for inter-jurisdictional registration for nurses in two Canadian provinces was just beginning at the time of our review [76].

In the United States, the state-based system of occupational licensure has created a confusing web of virtual care requirements across health professions. For example, the Federation of State Medical Boards regularly updates its information on states that have modified their licensure requirements to support virtual care in response to the pandemic. At the time of our review, their document on this topic included 22 pages of detailed information on the regulatory variations between states [19]. A requirement that health professionals follow the standards of care of multiple states to provide virtual care out-of-state raises liability risks and potentially stifles broader use [51]. In the United States, like in Canada, this variability has been the subject of debate and calls for national regulation of virtual care standards of practice (justified through federal powers over commerce and spending). Several papers identified and discussed the development of national licensure systems and licenses as a possible solution to the barriers and inefficiencies imposed by a state licensure system [65,66,77,78]. Achieving this would require the successful navigation of many potential political and legal difficulties [66,79]. However, some exceptions to state jurisdiction over licensure currently exist. Some federal health programs permit out-of-state clinicians to provide virtual care services within the Veterans Affairs system [65], perhaps due to the salaried nature of Veterans Affairs health providers. While not the focus of our review, the varying models of reimbursement for health services across state borders and internationally, particularly the common fee-for-service model, complicate moves to standardize virtual care, another potential barrier to broader implementation.

The literature demonstrated the value of policy harmonization and cooperation in the global digital economy. Canada and the United States can look to other countries and regions to see how they have adapted their occupational licensure systems and embraced new technologies for professional practice in the 21st century. For example, Australia has a national agency responsible for the regulation and accreditation of 15 health professions [80]. The European Economic Community recognizes sectoral professions including medicine, dentistry, nursing, pharmacy, and midwifery of member states to facilitate labour mobility, and the EU has a

system of regulation which automatically recognizes professional qualifications, thereby allowing credentialed professionals to lawfully practice in any member state [81].

**5. The demand for virtual care during COVID-19 catalyzed licensure and scope of practice changes**

The expansion of virtual care is anticipated to be an enduring legacy of COVID-19 in a pandemic recovery and post-pandemic world [43]. Regulatory reforms proliferated during the COVID-19 pandemic as regulatory bodies sought to ensure access to professional services while maintaining public protection. Grey literature reports described the rapid regulatory changes and waivers used to enable cross-border practice for virtual care, and the value of flexible rather than directive legislation to promote these types of regulatory responses [82–84]. For example, World Physiotherapy noted the various regulatory changes needed by physiotherapy regulators in different global jurisdictions to facilitate cross-border practice and provide guidance to registrants on standards for telerehabilitation services [84]. In the United States, executive orders from state governors temporarily lifted licensing requirements to solve workforce shortages related to an emergency. According to the National Conference of State Legislatures, before the start of the COVID-19 pandemic in 2020, only three governors had used executive orders for this purpose [67]. During the pandemic, all states took some sort of action to ease licensing requirements and meet workforce needs caused by the pandemic (most acutely seen in health and emergency response occupations). During the pandemic, legislation was also used to permit pharmacists, among other credentialed providers, to provide broader virtual care services to clients with fewer restrictions [85]. There was also a stronger push for states to join licensure compacts. A bill was introduced in the United States Congress in November 2020 that proposed penalizing states that did not join the IMLC and blocking state medical regulators from receiving specific federal grants if they did not have a public awareness campaign encouraging specialist physicians to practice virtual care [86].

## Discussion

The concept of regulating in the public interest continues to evolve in relation to virtual care. The emergence of the digital economy and the rapid shift to virtual service provision seen during the COVID-19 pandemic accelerated some of this evolution, with a shift from justifying virtual professional service provision to justifying any need for in-person requirements. The public interest is not just about quality service, as equitable accessibility and affordability are also in the interests of the public. Our review found a focus in the academic literature on balancing public safety with equitable access to services and economic competitiveness to determine the public interest [87]. This focus may continue to shift as we move into a post-pandemic world [88,89]. In particular, the regulatory implications of cross-jurisdictional virtual care provision and increasing privatization of virtual care have continued to be debated [90–93].

Related to our findings in this review around access to politically controversial health services and cross-jurisdictional practice, the rapidly changing legal landscape in the United States for reproductive health services raises new and important regulatory considerations for virtual care providers. The recent United States Supreme Court decision in *Dobbs v. Jackson Women's Health Organization* overturned the landmark decisions *Roe v. Wade* and *Planned Parenthood of Southeastern Pennsylvania v. Casey et al.* and removed the right to abortion in 26 states [94]. With the enactment of trigger laws following the *Dobbs* decision in several states [94,95], providers must understand any regulatory (e.g., licensure, documentation, confidentiality, accountability) impacts on reproductive services provided by virtual care. These impacts may include states revoking licensure compacts or other licensing reciprocity agreements,

exposure to litigation or criminal charges for assisting residents in obtaining abortion by virtual care, and the ability to prescribe or dispense abortion medication via virtual care [96]. Health profession regulators have a role in ensuring that registrants can access up-to-date standards and guidelines to understand any regulatory impacts of these new legal restrictions on abortion, including when the services are being provided through virtual care modalities.

Professional regulators may need to grapple with other emerging virtual practice issues more proactively to keep pace with the changes in this area and provide fit-for-purpose guidance to practitioners. For example, artificial-intelligence-enabled practice and disruptive technologies have regulatory implications and will likely have increasing relevance in the coming years, especially with the accelerated adoption of health-related technology during the pandemic. Some of this technology became incorporated into practice very swiftly in direct response to immediate user needs, at times without appropriate regulation or clear evidence of benefit [97,98].

Professional regulators are essential in adapting to new technologies that impact practice. Standards and guidelines should include what is required for competent practice in the era of virtual service provision. Many regulators of legal professionals in the United States and Canada have added a duty of technological competence in order to meet the standards of competence for the profession [99,100]. Health profession regulators should consider whether such a technological competence requirement should be adopted for their own professions if the quality of service requires a certain level of technical understanding and use.

As jurisdictions begin to focus on pandemic recovery, health workforce shortages continue to strain healthcare systems globally. Regulators are considering different ways that digital advancements in healthcare could be actively encouraged while still being safely incorporated into the public domain. One of these options is a regulatory sandbox, a "safe space" established and overseen by a particular regulatory body that allows live testing of new services or models in a controlled way prior to implementing them in the wider sector [101]. Contrary to the common view of regulation as a barrier to technological advancement and product testing, regulatory sandboxes are intended to use regulation as support for ensuring responsible innovation [101]. More common in legal and financial technology regulation, the possibility of using regulatory sandboxes for testing and assessing virtual health initiatives has recently received academic attention [98,101]. Taking such an approach to different areas of regulation can send a positive message that regulators are open to new ways of meeting the public's needs, but this must be balanced against the sustainability of regulatory sandboxes and the ability of regulators to assess and monitor the quality of innovative virtual service development. Professional regulators have a role in regulating technologically-based care, but this regulation may require a new regulatory design that focuses less on individual professionals and more on desired outcomes [102].

Our scoping review indicated that access to healthcare was emphasized when discussing the concept of the public interest in regulating health professionals providing virtual care. However, populations with more significant access barriers—such as those living in rural or remote areas, people with disabilities, and people with low health literacy—can be even more marginalized by the increased adoption of virtual care [103,104]. Policymakers and healthcare stakeholders, including regulators, should act collaboratively to build virtual care services that consider inequities and promote equitable access to virtual healthcare [103,104].

## Limitations

Much of the literature involved descriptive case examples, critical commentaries, or legal analysis and predominantly focused on licensure in high-income Anglophone countries and the

historically dominant medical profession. It is likely that virtual care in the statutorily regulated health professions in these jurisdictions is more frequently highlighted in the literature. More publications from low- and middle- income countries, occupations outside medicine, and evidence published in other languages may have informed this review had they been included. Also, as most included academic literature was set in the American context, the findings may have differed if more articles from countries with universal health coverage had been included, limiting the generalizability of the findings. We only included sources since 2015 and did not include guidelines or documents from individual regulators. The search was conducted almost two years ago and there could be new evidence not included in this review.

Despite these limitations, this scoping review has provided a foundational base for conceptualizing the public interest when regulating health professionals providing virtual care. While the focus of a scoping review is to provide breadth rather than depth, insights from this review may be applied to inform regulatory approaches for health professional virtual care in steady state and in response to future emergencies.

## Future research

This scoping review has highlighted knowledge gaps in how health profession regulators can act in the public interest when regulating practitioners who provide virtual or digitally-enabled services. As most of the articles available were reviews or commentaries, there appears to be little empirical evidence on which to base these regulatory activities or to understand the impact of regulation on equitable access to and affordability of virtual care. Future research focused on outcomes is needed, as well as studies on low- and middle-income countries and health practitioners that are not regulated by statute (including the assistant, support, and allied health workforces).

The literature did not always touch on the public interest explicitly; hence there is a need for more research on the public interest in regulating virtual care in the health professions. There is also a need to understand how the public can best be engaged in the regulatory process, particularly around understanding the public interest when receiving virtual care and how this should translate into regulatory policies and practices. Reforms in recent years have included more inclusive and expansive participation of the public in regulation [4,105]. Public engagement with regulation for particular technologies may influence regulators to understand what initial assumptions about impact, harm, and risk may not be substantiated and allow for continual evaluation, adjustment and improvement [22]. Future work could partner with regulators, policymakers, health practitioners, and the public to enrich findings.

Many articles focused on the regulatory changes necessitated by the COVID-19 pandemic. It is imperative to continue to monitor and evaluate professional regulation through research to determine the value of these changes in the post-pandemic world. Future research should assess the outcomes of these regulatory initiatives, with a specific lens on accessibility and equity aspects of the public interest.

## Conclusions

Virtual care has the potential to improve the quality and equity of services but may also introduce new areas of potential risk and amplify existing inequalities, particularly with the digital divide that exists along socioeconomic and geographic lines [58]. Regulators should consider the impact of these equity and access issues on the public interest when providing guidance to health professionals on providing virtual care. Clear standards and guidelines for virtual care are needed, including requirements for competent practice since traditional definitions may be outdated for digitally enabled service provision.

While the literature did not always explicitly discuss the public interest, the reviewed literature focused on balancing public safety with equitable access to services and economic competitiveness, particularly given the pandemic-driven demand for virtual care. The literature also identified subnational licensure as a barrier to equitable access to virtual care, reflecting the dominance of federated countries in the reviewed literature. This concern and other aspects of cross-jurisdictional practice remain critical to regulating professionals providing virtual care in these jurisdictions and will continue to be relevant given global health workforce challenges. To leverage virtual care effectively both in steady state and in future global public health emergencies, governments and regulators should work together to facilitate virtual care provision across jurisdictional borders and standardize the regulation of virtual care in the public interest.

## Supporting information

**S1 PRISMA-ScR Checklist. Preferred Reporting Items for Systematic Reviews and Meta-Analyses extension for Scoping Reviews (PRISMA-ScR) checklist.**
(PDF)

**S1 File. Search strategy for MEDLINE via Ovid conducted on 29 May 2021.**
(DOCX)

## Acknowledgments

We gratefully acknowledge the valuable research assistance of Maggie Szu Ning Lin and Aleah McCormick. We also thank academic librarian Nicole Askin for her assistance on the search strategy, Sarah Stahlke for her thoughtful review on a previous version of the manuscript, and the three peer reviewers for their constructive and helpful insights to improve the manuscript.

## Author Contributions

**Conceptualization:** Kathleen Leslie.

**Formal analysis:** Kathleen Leslie, Sophia Myles, Catharine J. Schiller, Sioban Nelson, Tracey L. Adams.

**Funding acquisition:** Kathleen Leslie.

**Investigation:** Kathleen Leslie, Sophia Myles, Catharine J. Schiller, Sioban Nelson, Tracey L. Adams.

**Methodology:** Kathleen Leslie.

**Project administration:** Kathleen Leslie.

**Writing – original draft:** Kathleen Leslie, Sophia Myles, Catharine J. Schiller, Abeer A. Alraja, Sioban Nelson, Tracey L. Adams.

**Writing – review & editing:** Kathleen Leslie, Sophia Myles, Catharine J. Schiller, Abeer A. Alraja, Sioban Nelson, Tracey L. Adams.

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
