## [Decision Letter · Decision Letter 0]

6 Feb 2023

PDIG-D-22-00332

Protecting the public interest while regulating health professionals providing virtual care: A scoping review

PLOS Digital Health

Dear Dr. Leslie,

Thank you for submitting your manuscript to PLOS Digital Health. After careful consideration, we feel that it has merit but does not fully meet PLOS Digital Health's publication criteria as it currently stands. Therefore, we invite you to submit a revised version of the manuscript that addresses the points raised during the review process.

Please submit your revised manuscript within 30 days Mar 08 2023 11:59PM. If you will need more time than this to complete your revisions, please reply to this message or contact the journal office at digitalhealth@plos.org. Please include the following items when submitting your revised manuscript:

We look forward to receiving your revised manuscript.

Kind regards,

Baki Kocaballi

Academic Editor

PLOS Digital Health

Journal Requirements:

Additional Editor Comments (if provided):

We are pleased to inform you that your paper submission has been accepted for publication with minor revisions. The reviewers were impressed with the quality of your work and the valuable contributions it makes to the field. They have made some useful suggestions that will help further strengthen the paper. Please take the time to carefully review and address the reviewers' comments before publication.

Reviewers' comments:

Reviewer's Responses to Questions

**Comments to the Author**

1. Does this manuscript meet PLOS Digital Health’s publication criteria? Is the manuscript technically sound, and do the data support the conclusions? The manuscript must describe methodologically and ethically rigorous research with conclusions that are appropriately drawn based on the data presented.

Reviewer #1: Yes

Reviewer #2: Yes

Reviewer #3: Yes

2. Has the statistical analysis been performed appropriately and rigorously?

Reviewer #1: Yes

Reviewer #2: N/A

Reviewer #3: Yes

3. Have the authors made all data underlying the findings in their manuscript fully available (please refer to the Data Availability Statement at the start of the manuscript PDF file)?

Reviewer #1: Yes

Reviewer #2: Yes

Reviewer #3: Yes

4. Is the manuscript presented in an intelligible fashion and written in standard English?

Reviewer #1: Yes

Reviewer #2: Yes

Reviewer #3: Yes

5. Review Comments to the Author

Reviewer #1: Please see attachment for all of my comments.

I have two recommendations regarding the framing/presentation of the results to make them more accessible and also so that they better conform with the existing literature.

Reviewer #2: Thank you for the opportunity to review this interesting manuscript. It is well written and original and of interest to the community of regulators and regulated and those that study these areas. I have a few minor comments that can be considered as nuancing the paper to enhance its findings rather than any major critique.

At line 46 the authors state "Professional regulators are legally obligated to protect the public interest, but how the public interest is defined is amorphous and subject to social change". I of course agree with this but in light of future discussion in the paper and the intent to discuss literature from across a variety of jurisdictions I suggest also noting in this sentence that what constitutes public interest in regulation is also affected by the state/province, country or supra national entity in which regulation takes place. See for example Feintuck The Public interest in Regulation (OUP). Given the range of countries included stretch from Russia to the US this is important to clarify.

Line 56 discusses high income Anglophone nations - generally most of the materials analysed fall within this cohort but not all. Additionally, many LMIC nations also regulate in the public interest but have different priorities and resourcing and framing of the public interest. Also relevant is whether the nation considers health care a public or private good. Many nations have or are working towards universal health systems recognising health as a public good - the US is an outlier in the OECD in its framing of health care as predominantly a private good.

223 while the authors note that most of the literature did not focus on the public interest. I wonder if this a framing issue rather than a lack of focus as, as noted by the authors, this was implicit rather than explicit. Thus I wonder whether this section should be focused on a deficit framing of not focusing on public interest or a more nuanced framing that considered implicitly? This would also be better aligned with the aforementioned amorphous nature of the public interest and how it might be quite jurisdictionally specific in its construction.

Finding 3 (line 296). The literature included for this review was dominated by literature from the US - where competition and politics in health care are significant issues due to the political system of that country and I would suggest its framing of health as predominantly a private good and/or individual responsibility. Although there is a one sentence acknowledgement that competition might bear a different weight in the US versus other nations the same could be said for politics. In some nations the discussion is how to enable assessment for abortion, assisted dying etc to be undertaken virtually to enable access to what are clearly framed as health services (public goods) in those nations (an access argument). The generalisability of this finding as a whole, outside the US context, needs to be much more clearly and explicitly nuanced as part of the conclusion to this section.

Finding 4 (line 341) The nature of cross-national reviews of legislation is complex. While issue 4 is a big one for federal states, there are unitary states (eg New Zealand etc) where regulation has always occurred nationally. For this group of countries cross-jurisdictional virtual practice is not a thing but regulating other aspects of the public interest around safety etc is the focus for any regulation around virtual care. This section needs to be nuanced to reflect this. 

Line 452 please define regulatory sandbox and add a reference. It would be difficult for readers who lack knowledge of what this means and how and why it works to follow why this might be a beneficial approach in this context without further information. 

The conclusion and discussion section should also be nuanced per the comments above. While there is a clear statement of limitations in the conclusion these limitations and the lack of generalisability of some of the findings needs to be clearer throughout. 

Just for your interest, another example of legal barriers to providing virtual care comes from Australia where currently discussions between health professionals and patients around assisted dying cannot be conducted over a carriage service due to federal law. See for example Del Villar, Katrine, Close, Eliana, Hews, Rachel, Willmott, Lindy, & White, Ben (2022) Voluntary assisted dying and the legality of using a telephone or internet service : The impact of Commonwealth ‘Carriage Service’ offences. Monash University Law Review, 47(1), pp. 125-173.

Reviewer #3: The opportunity to review this contribution for the PLOS Digital Health Journal is greatly appreciated. The use of virtual technology for healthcare has the potential to reduce a variety of challenges associated with accessing healthcare services for remote populations and to improve the quality of care. As a result, this scoping review will provide evidence-based information regarding virtual healthcare. However, I have minor suggestions:

1. Abstract: the authors should include the sources of data under the methods section, even though they mentioned six multidisciplinary databases. The name of the database used to search the studies should be mentioned in the methods section.

2. Introduction: In the introduction of the study, the authors have very clearly explained the rationale and significance of this particular study. However, the introduction is a little bit long and it would be better if the authors reduced it one and a half pages.

3. Methods: Authors should report in Table the keywords or search strings that they used to identify relevant studies during the search strategy. 

4. The paper is generally well-written, and I strongly recommend that it be published.

6. PLOS authors have the option to publish the peer review history of their article (what does this mean?). If published, this will include your full peer review and any attached files.

**Do you want your identity to be public for this peer review?** For information about this choice, including consent withdrawal, please see our Privacy Policy.

Reviewer #1: Yes: Edward J. Timmons

Reviewer #2: No

Reviewer #3: Yes: NITTARI G.

---

## [Editor Report · Decision Letter 1]

20 Mar 2023

Protecting the public interest while regulating health professionals providing virtual care: A scoping review

PDIG-D-22-00332R1

Dear Dr Leslie,

We are pleased to inform you that your manuscript 'Protecting the public interest while regulating health professionals providing virtual care: A scoping review' has been provisionally accepted for publication in PLOS Digital Health.

Many thanks for your revisions. The revised version of the paper has addressed the reviewers' comments satisfactorily. I recommend including the search strategy as an appendix. Please also include a statement in the limitations section that the search was done almost 2 years ago and there could be some new evidence available that is not covered in the current review.

Before your manuscript can be formally accepted you will also need to complete some formatting changes, which you will receive in a follow-up email from a member of our team. 

Best regards,

Baki Kocaballi

Academic Editor

PLOS Digital Health

Padmanesan Narasimhan

Section Editor

PLOS Digital Health